# Deep Kernel Learning of Nonlinear Latent Force Models

**Jacob Moss**                                                                  *jm2311@cam.ac.uk*
*Computer Lab*
*University of Cambridge*

**Jeremy England**
*GSK*
*Petach Tikva, Israel*

**Pietro Lió**
*Computer Lab*
*University of Cambridge*

**Reviewed on OpenReview:** *https://openreview.net/forum?id=CNJIpI4Gb9*

## Abstract

Scientific processes are often modelled by sets of differential equations. As datasets grow, individually fitting these models and quantifying their uncertainties becomes a computationally challenging task. Latent force models offer a mathematically-grounded balance between data-driven and mechanistic inference in such dynamical systems, whilst accounting for stochasticity in observations and parameters. However, the required derivation and computation of the posterior kernel terms over a low-dimensional latent force is rarely tractable, requiring approximations for complex scenarios such as nonlinear dynamics. In this paper, we overcome this issue by posing the problem as learning the solution operator itself to a class of latent force models, thereby improving the performance and scalability of these models. This is achieved by employing a deep kernel along with a meta-learned embedding of the output functions. Finally, we demonstrate the ability to extrapolate a solution operator trained on simulations to real experimental datasets, as well as scaling to large datasets.

## 1 Introduction

Differential equations are mathematical models that describe the change of a function with respect to one or more variables, such as time. They play a central role in the natural and social sciences, providing a way to model and understand complex systems and phenomena, such as the growth and decline of populations (Burghes, 1975), morphogenesis (Turing, 1990), the dynamics of biochemical reactions (Thomas et al., 1976; Schoeberl et al., 2002; Barenco et al., 2006), and so on. They provide a rigorous, well-studied, and mathematically grounded method of making historic and future predictions in complex systems. In a machine learning context, the modelling power of differential equations make them excellent inductive biases. A popular method of incorporating these equations within a Bayesian machine learning setting is the latent force model (LFM), introduced by Lawrence et al. (2006) to model a network of genes regulated by a common protein. LFMs assume that the underlying dynamics of a system can be modelled in terms of a low-dimensional latent force, typically with a Gaussian process prior, within a system of differential equations. The involvement of a non-parametric Gaussian process within an interpretable parametric system results in a powerful framework for drawing mechanics-constrained inferences in noisy, high-dimensional, and nonlinear dynamics. It has since been extended to embryogenesis (López-Lopera et al., 2019), where the latent force represents mRNA concentration; patient interventions (Cheng et al., 2020), with latent forces corresponding to different treatments; and movement segmentation (Alvarez et al., 2010b). However, there are significant computational challenges hindering the usability of these models on large datasets.

Inferring the latent force requires computing the posterior distribution, the covariance functions of which are determined by the differential equations describing the model dynamics. These are analytically tractable only for a small set of scenarios. The remaining cases, typically nonlinear dynamical systems, requires some approximation for the posterior. There are various approximations, such as using an ODE or PDE solver (Moss et al., 2021), deep GPs (McDonald & Álvarez, 2021), and filtering (Hartikainen & Sarkka, 2012; Ward et al., 2020). Crucially, however, these existing works operate on small datasets—often a single parameterisation—not considering the scaling of the approach to multiple independent tasks of the same LFM. How do we extend the original model considering one network of genes to many thousands? Currently, important use-cases such as fitting multiple ODEs (Santra, 2018) or performing Bayesian model selection in dynamical models (Babtie et al., 2014) with LFMs are too costly. Increasingly large datasets are only underlining the importance of solving the challenge of scaling up LFMs to work in the multi-task setting.

In this paper, we propose a novel method addressing these scalability issues, resulting in 250x to 15,000x faster latent force inferences when compared with existing approximations. We propose a deep learning approach to solve a general class of LFMs, avoiding differential equation solving steps and variational approximations by instead learning the dynamics in a deep kernel (Wilson et al., 2016). Our framework makes use of neural network representations of sets, such as the Transformer (Vaswani et al., 2017) and the Fourier neural operator (Li et al., 2020), in order to produce a function embedding for each task. Given a task consisting of only the input mesh, for example time, and the observed functions' embedding, our model infers the associated latent force with Gaussian process conditioning. This makes our approach much faster than training an LFM on individual tasks. This method can model complex nonlinear dynamics and provides solutions even to multivariate problems such as partial differential equations which were previously computationally infeasible for large datasets.

## 2 Preliminaries

**Gaussian processes**   Gaussian processes are stochastic processes often used as priors for latent functions in Bayesian machine learning models that map from inputs $\boldsymbol{x} \in \mathbb{R}^D$ to predictions $f(\boldsymbol{x}) \in \mathbb{R}$. A GP prior

$$f \sim \mathcal{GP}(m(\boldsymbol{x}), \kappa(\boldsymbol{x}, \boldsymbol{x}')) \tag{1}$$

is described by its mean function $m(\boldsymbol{x})$ and its kernel function $\kappa(\boldsymbol{x}, \boldsymbol{x}')$. The mean function is usually set to 0 for standardised data. The kernel function may have a set of hyper-parameters $\theta$, such as the lengthscale $l$ in an RBF kernel, $k_{\text{RBF}}(\boldsymbol{x}, \boldsymbol{x}') = \exp(-\frac{l}{2}\|\boldsymbol{x} - \boldsymbol{x}'\|_2)$. Under this prior, any finite collection of points $\mathbf{f}(\boldsymbol{X})$ for inputs $\boldsymbol{X} = [\boldsymbol{x}_1, \boldsymbol{x}_2, \ldots, \boldsymbol{x}_N]^\top$ is normally distributed: $\mathbf{f} \sim \mathcal{N}(m(\boldsymbol{X}), \kappa(\boldsymbol{X}, \boldsymbol{X}))$. In this paper, we denote the covariance $\boldsymbol{K}_{ff} = \kappa(\boldsymbol{X}, \boldsymbol{X})$. With a Gaussian likelihood for observations $y$, meaning $y \sim \mathcal{N}(f, \sigma^2)$, the posterior distribution for training data $\boldsymbol{X}, \boldsymbol{y}$ is analytically tractable and given by

$$f \mid \boldsymbol{y} \sim \mathcal{N}\bigg(\kappa(\boldsymbol{x}, \boldsymbol{X})[\kappa(\boldsymbol{X}, \boldsymbol{X}) + \sigma^2 \mathbf{I}]^{-1}\boldsymbol{y}, \kappa(\boldsymbol{x}, \boldsymbol{x}') - \kappa(\boldsymbol{x}, \boldsymbol{X})[\kappa(\boldsymbol{X}, \boldsymbol{X}) + \sigma^2 \mathbf{I}]^{-1}\kappa(\boldsymbol{X}, \boldsymbol{x}')\bigg).$$

Moreover the marginal likelihood has a closed form expression enabling optimising the kernel hyper-parameters $\theta$ by maximising the marginal likelihood using gradient-based optimisation and is given by

$$p(\boldsymbol{y}) = \mathcal{N}(\boldsymbol{y} \mid \mathbf{0}, \kappa(\boldsymbol{X}, \boldsymbol{X}) + \sigma^2 \mathbf{I}). \tag{2}$$

**Deep Kernel Learning**   Deep kernel learning as presented by Wilson et al. (2016) constitutes an attempt to combine the representation learning capabilities of deep neural networks with the non-parametric nature of Gaussian processes. A neural network is used to map an input $\boldsymbol{x}$ into a latent space yielding a vector $\text{NN}(\boldsymbol{x}) \in \mathbb{R}^D$. This representation is then fed into a base kernel $\kappa(\cdot, \cdot)$ (such as an RBF kernel) to yield the covariance between inputs $\kappa(\text{NN}(\boldsymbol{x}), \text{NN}(\boldsymbol{x}'))$.

**Latent Force Models**   LFMs incorporate explicit dynamics of differential equations in the kernel functions of Gaussian processes (GPs) in order to infer latent forcing terms (Lawrence et al., 2006; Alvarez et al., 2009). The latent force captures the underlying process and structure in the data, while being unobserved

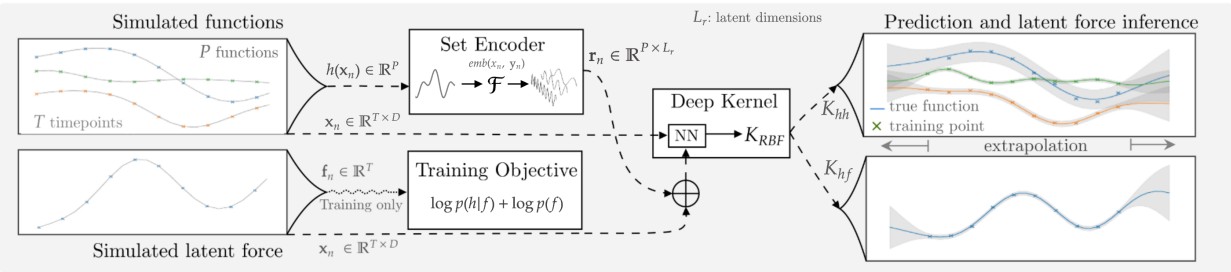

Figure 1: Schematic of DKLFM. First, a dataset of latent force tasks is created by sampling the latent force and differential equation parameters and solving the forward solution. The simulated functions are embedded by aggregating the output state of an encoder. A deep kernel is learned to represent the convolution operator of an arbitrary LFM. For training tasks, the model minimises the loss in Equation 6 with access to simulated latent force data. For test tasks, the latent force is unobserved and inferred via the cross-covariance only, as in a typical LFM scenario. The diagram shows one task; in reality, we train over batches of tasks.

and shared amongst the outputs. The differential equation, $g$, parameterised by $\boldsymbol{\Theta}$, enforces a mechanistic relationship between the vector-valued output function, $h(\boldsymbol{x}) \in \mathbb{R}^P$, and an unobserved, scalar-valued latent force, $f(\boldsymbol{x}) \in \mathbb{R}$, with $D$-dimensional input $\boldsymbol{x} \in \mathbb{R}^D$. This input is often time but can be any variable. An ordinary differential equation (ODE) involves derivatives of a function of one independent variable, describing how this function changes ($D = 1$). A partial differential equation (PDE) involves derivatives with respect to multiple variables, used to model multi-dimensional phenomena like heat and fluid dynamics ($D > 1$).

$$\frac{\mathrm{d}h(\boldsymbol{x})}{\mathrm{d}\boldsymbol{x}} = g_{\boldsymbol{\Theta}}\big(\boldsymbol{x}, \boldsymbol{h}, G(f(x))\big), \tag{3}$$

or equivalently for a PDE. The force can be transformed by some response function $G(\cdot)$. A GP prior is assigned to the latent force, $f \sim \mathcal{GP}(\boldsymbol{0}, \kappa(\boldsymbol{x}, \boldsymbol{x}'))$, which naturally accounts for biological noise and enables non-linear expressivity through kernels. Some LFM literature considers multiple forces but we do not cover this due to the identifiability issues they pose. An analytical expression for the covariance between outputs, $\boldsymbol{K}_{hh}$, is possible under the necessary condition that $G$ is a linear operator. In these cases, maximum marginal likelihood yields the differential equation parameters and inference can be carried out with standard posterior GP identities (see Rasmussen & Williams (2005)). Approximations are required where $G$ is non-linear.

## 3 Deep Kernel Learning of Latent Force Models

In this section, we present the Deep Kernel Latent Force Model (DKLFM): a novel approach to multi-task dynamical modelling. We first detail the problem setting and derive our objective function, and finally discuss any design choices in our approach.

### 3.1 Model Formulation

The model setup is summarised in Figures 1 and 2, illustrating the generative process. Our input $x$ is transformed by the latent force $f$, which is further manipulated by a set of differential equations to yield the output functions $h$. We aim to solve the inverse problem of inferring $f$ given noisy observations of $h$. We start by illustrating our approach with a model used later in this paper (Barenco et al., 2006). The time derivative of the mRNA, $h(\boldsymbol{x})$, for $P$ genes is related to its regulating transcription factor protein, $f(x)$ by

$$\frac{\mathrm{d}h(\boldsymbol{x})}{\mathrm{d}\boldsymbol{x}} = g_{\boldsymbol{\Theta}}\big(\boldsymbol{x}, \boldsymbol{h}, G(f(x))\big) = \overbrace{\boldsymbol{b}}^{\text{basal rate}} + \boldsymbol{s}\overbrace{G(f(\boldsymbol{x}))}^{\text{response}} - \overbrace{\boldsymbol{c}\,h(\boldsymbol{x})}^{\text{decay term}}, \tag{4}$$

where $\boldsymbol{x} \in \mathbb{R}^1$ is time and $\boldsymbol{b}, \boldsymbol{s}, \boldsymbol{c} \in \mathcal{R}_+^P$ are the base transcription rates, sensitivities to the transcription factor, and decay rates of the $P$ genes respectively. $G$ is a function, for example a nonlinearity enforcing positivity or a saturation term enforcing limits on the latent force. The ODE parameters are thus $\boldsymbol{\Theta} = \{\boldsymbol{b}, \boldsymbol{s}, \boldsymbol{c}\}$.

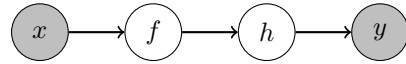

Figure 2: Graphical model illustrating the generative process from latent forces to output functions.

At training time, we assume a simulated dataset of $N$ tasks, each consisting of a different latent force function and setting of parameter values. The $n$-th task is the set $\{\boldsymbol{X}_n, \boldsymbol{Y}_n, \boldsymbol{f}_n\}$, where $\boldsymbol{X}_n \in \mathbb{R}^{T \times D}$ denotes $T$ observed $D$-dimensional input points, e.g. temporal ($D = 1$) and spatio-temporal ($D > 1$). We have noisy observations, $\boldsymbol{Y}_n \in \mathbb{R}^{T \times P}$, of the $P$-dimensional solutions to the differential equation at each of these $T$ input points. The solutions are assumed to follow the GP $h$ with realisations $\boldsymbol{h}_n$ where the $P$ dimensions are flattened. The observations $\boldsymbol{Y}_n$ are similarly condensed into the blocked vector $\boldsymbol{y}_n$. Since we carry out training on a simulated dataset generated by the LFM, we also have access to latent force observations, $\boldsymbol{f}_n \in \mathbb{R}^T$. For clarity, we henceforth omit the subscript task index and it can be assumed that we are dealing with each task independently. These tasks are split into train and test sets. At inference time, our objective is to infer the latent force and the model makes use only of the output function observations.

We assign a GP prior to $f$ and use a Gaussian likelihood, i.e.

$$f \sim \mathcal{GP}(m_f(\cdot), \kappa_f(\cdot, \cdot)), \qquad \text{(prior)}$$
$$y \sim \mathcal{N}(\boldsymbol{h}, \sigma^2 \boldsymbol{I}), \qquad \text{(likelihood)}$$

where $m_f$ is the mean function of the GP prior for which we learn a constant output, and $\kappa_f$ is the kernel.

For each task, the distribution over output realisations $\boldsymbol{h}$ is implicitly determined via their joint distribution with the latent function outputs, $\boldsymbol{f}$. In practice we make a modelling assumption that this joint is multivariate Gaussian. Inferences are then made using GP conditioning on the output observations for test tasks. This results in the joint distribution

$$\begin{bmatrix} \boldsymbol{f} \\ \boldsymbol{h} \end{bmatrix} \sim \mathcal{N}\left( \begin{bmatrix} \boldsymbol{\mu}_f \\ \boldsymbol{\mu}_h \end{bmatrix}, \begin{bmatrix} \boldsymbol{K}_{ff} & \boldsymbol{K}_{fh} \\ \boldsymbol{K}_{hf} & \boldsymbol{K}_{hh} \end{bmatrix} \right), \qquad \text{(joint)}$$

where the mean vectors and covariance matrices are obtained as described below. This assumption of a multivariate Gaussian joint distribution holds only for linear transformations of the latent force (i.e. where $G$ is a linear function). Nevertheless, we demonstrate in Section 4.3 that this approximation yields strong quantile similarities to the true empirical distribution even for nonlinear transformations.

**Deep kernels** In an LFM, the kernel function is derived from solving a set of differential equations. In the case of a general non-linear equation, the kernel has no closed-form solution. We instead approximate the kernel with a neural network of sufficient capacity, mapping the inputs to latent representation vectors of size $L_d$ before feeding them into a base kernel, $\kappa : \mathbb{R}^{L_d} \times \mathbb{R}^{L_d} \to \mathbb{R}$ (Wilson et al., 2016). This base kernel provides an additional inductive bias, for example ensuring smoothness in the latent space with an RBF kernel or periodicity with a periodic kernel. To map to this latent space, we construct two separate networks $\text{NN}_f : \mathbb{R}^{D+L_r} \to \mathbb{R}^{L_d}$, $\text{NN}_h : \mathbb{R}^{D+L_r} \to \mathbb{R}^{L_d}$ for the latent and output functions respectively, where $L_r$ is the size of the task representation, $\boldsymbol{r} \in \mathbb{R}^{P \times L_r}$, discussed further in the next section. We also need a common network, $\text{NN}_c : \mathbb{R}^{L_d} \to \mathbb{R}^{L_d}$, which maps both representations onto the same latent space. This common network helps to obtain an informative cross-covariance between latent force and output functions: we need to map both to a common latent space. For simplicity, we select a simple MLP for all networks. We found that incorporating skip connections greatly improved the performance whilst reducing overfitting. To illustrate for task $n$, the cross-function covariances, $K_{fh}$ and $K_{hh}$, are computed as follows. The deep kernel receives a concatenation of the task representation with the inputs, $\boldsymbol{z}_n = \boldsymbol{r}_n \oplus \boldsymbol{x}_n$,

$$\boldsymbol{K}_{fh} = \kappa\left(\text{NN}_c(\text{NN}_f(\boldsymbol{z}_n)), \text{NN}_c(\text{NN}_h(\boldsymbol{z}'_n))\right), \quad \boldsymbol{K}_{hh} = \kappa\left(\text{NN}_c(\text{NN}_h(\boldsymbol{z}_n)), \text{NN}_c(\text{NN}_h(\boldsymbol{z}'_n))\right). \qquad (5)$$

The same concatenation occurs for the latent covariance $\boldsymbol{K}_{ff}$ and for the mean functions, e.g. $\boldsymbol{\mu}_f = m_f(\boldsymbol{z}_n)$. The covariance matrices are therefore of shape $\boldsymbol{K}_{ff} \in \mathbb{R}^{T \times T}$, $\boldsymbol{K}_{fh} \in \mathbb{R}^{T \times T \cdot P}$, and $\boldsymbol{K}_{hh} \in \mathbb{R}^{T \cdot P \times T \cdot P}$.

In some experiments, particularly periodic scenarios, it helped to concatenate the input mesh after applying the neural network. This granted the periodic kernel access to both the latent vector and input mesh. For example, based on Equation 5, $K_{fh} = \kappa_{\text{periodic}}\left(\boldsymbol{x}_n \oplus \text{NN}_c(\text{NN}_f(\boldsymbol{z}_n)), \boldsymbol{x}_n \oplus \text{NN}_c(\text{NN}_h(\boldsymbol{z}'_n))\right).$

The deep kernel weights and the base kernel and mean function hyperparameters are optimised jointly by maximising the marginal likelihood of the output functions and latent force. This has the closed form:

$$p(\boldsymbol{y}, \boldsymbol{f}) = \int p(\boldsymbol{y}, \boldsymbol{f}, \boldsymbol{h}) \, \mathrm{d}\boldsymbol{h} = \int p(\boldsymbol{y}|\boldsymbol{h})p(\boldsymbol{h}|\boldsymbol{f})p(\boldsymbol{f}) \, \mathrm{d}\boldsymbol{h}$$

$$= p(\boldsymbol{f}) \int \mathcal{N}(\boldsymbol{y} \,|\, \boldsymbol{h}, \sigma^2 \mathbf{I}) \mathcal{N}(\boldsymbol{h} \,|\, \boldsymbol{\mu}_{h|f}, \boldsymbol{K}_{h|f}) \, \mathrm{d}\boldsymbol{h}$$

$$= \mathcal{N}(\boldsymbol{f} \,|\, \boldsymbol{\mu}_f, \boldsymbol{K}_{ff}) \mathcal{N}(\boldsymbol{y} \,|\, \boldsymbol{\mu}_{h|f}, \boldsymbol{K}_{h|f} + \sigma^2 I). \tag{6}$$

where in practice the negative log marginal likelihood is minimised and

$$\log \mathcal{N}(\boldsymbol{y} \,|\, \boldsymbol{\mu}_{h|f}, \boldsymbol{K}_{h|f} + \sigma^2 I) = -\frac{1}{2}(\boldsymbol{y} - \boldsymbol{\mu}_{h|f})^T (\boldsymbol{K}_{h|f} + \sigma^2 I)^{-1}(\boldsymbol{y} - \boldsymbol{\mu}_{h|f}) - \frac{1}{2}\log|\boldsymbol{K}_{h|f} + \sigma^2 I|$$

Here, $\boldsymbol{\mu}_{h|f}$ and $\boldsymbol{K}_{h|f}$ are defined as

$$\boldsymbol{\mu}_{h|f} = \boldsymbol{\mu}_h + \boldsymbol{K}_{hf} \boldsymbol{K}_{ff}^{-1} \boldsymbol{f}, \tag{7}$$

$$\boldsymbol{K}_{h|f} = \boldsymbol{K}_{hh} - \boldsymbol{K}_{hf} \boldsymbol{K}_{ff}^{-1} \boldsymbol{K}_{fh}. \tag{8}$$

At inference time, we receive an arbitrary input $\boldsymbol{x}^*$. At these input locations, we define $\boldsymbol{f}^*$ as the inferred and unobserved latent force using the posterior predictive distribution, defined by

$$\boldsymbol{\mu}_{f*|y} = \boldsymbol{\mu}_{f*} + \boldsymbol{K}_{f*h} \boldsymbol{K}_{hh}^{-1} \boldsymbol{y}, \tag{9}$$

$$\boldsymbol{K}_{f*|y} = \boldsymbol{K}_{f*f*} - \boldsymbol{K}_{f*h} \boldsymbol{K}_{hh}^{-1} \boldsymbol{K}_{hf*}. \tag{10}$$

Note that we are using exact GP inference in this approximate model. If the input space is very large, then the matrix inversion can become computationally challenging and a variational approximation may be easily interchanged here. We will now discuss specific components of the model in more detail.

## 3.2 Task Representation

The generative process relies on both the latent force and a set of task-specific parameters relating to the dynamics equations, such as reaction and decay rates. By tweaking these parameters, it is possible that latent forces from different tasks could produce the same or similar output functions. Moreover, tasks on the same input mesh would result in identical covariance matrices, limiting cross-task utility. The GP therefore needs access to a task representation in order to generalise across tasks. We denote this $\boldsymbol{r}_n = \mathtt{emb}(\boldsymbol{x}_n, \boldsymbol{y}_n)$, defined by $\mathtt{emb} : \mathbb{R}^{P \times T} \to \mathbb{R}^{P \times L_r}$, where the embedding size, $L_r$, is a hyper-parameter and each of the $P$ output functions are treated independently. Note that it does not observe any latent force data. Instead, this representation is used by the deep kernel to learn the relationship between the output functions and the latent force in its latent space. This separates inference of latent forces from the dynamics parameters, whilst enabling the task representation to be computed for tasks where no latent force observations exist.

This embedding must contain the dynamics information usually captured by the differential equations and their parameters, and should be invariant to input resolution in order to maintain the flexibility of Gaussian processes. This also enables super-resolution inference; test cases can be at an arbitrary resolution higher than the training data, as we demonstrate in Section 4.2. To that end, we explored two different encoders in our research: a Fourier neural operator (Li et al., 2020) and a Transformer (Vaswani et al., 2017).

The Fourier neural operator is effectively an MLP in the Fourier domain, and was originally developed to solve partial differential equations (PDEs) due to the intricate relationship between differential calculus and Fourier analysis, their mesh-invariance, and fast solving speeds. However, a severe limitation is that the Fourier transform requires the input to be a regularly spaced. On the other hand, the Transformer is invariant to resolution and regularity of observations. We implemented the decoder only consisting of a linear layer followed by self-attention layers. Sinusoidal positional encoding enables the modelling of an irregular mesh. The weights are trained by maximising the marginal likelihood in Equation 6.

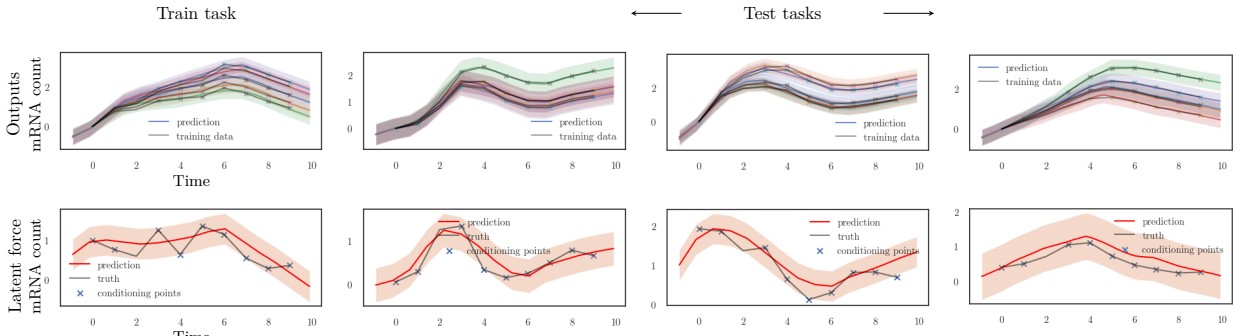

Figure 3: Training and test transcriptional regulation tasks. Notice that even for test tasks, the learned variance encapsulates most ground truth. Test tasks do not have access to latent force data.

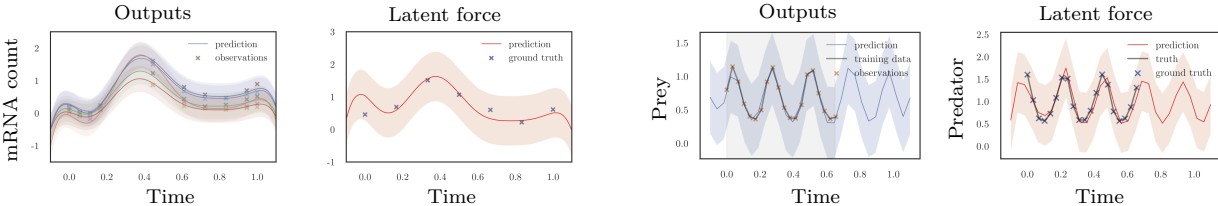

(a) DKLFM infers the protein concentration of transcription factor p53. The ground truth was published by Barenco et al. (2006). The model was trained only on simulations of Equation 4.

(b) DKLFM infers the predator-prey relationship in a Lotka-Volterra setup. The model has only been trained within the time range denoted by the grey shading, and extrapolates the periodic nature beyond.

Figure 4: DKLFM extrapolation in both tasks and input domain.

## 4    Experiments

In this section we investigate the performance of DKLFM on two ODE-based LFMs and one PDE-based LFM. Given that this is the first multi-task model for latent force models, we analyse the performance on real, experimentally-derived datasets not in the synthetic training distribution. We compare our approach with two models from the literature on solving latent force problems. *Alfi* is a variational approximation with a *strong* mechanistic prior, meaning the model is constrained by the differential equations. For linear $g$, *Alfi* resorts to the exact solution. The *DeepLFM* involves dynamics-informed random features which are composed with each layer. This results in only a *mid*-strength mechanistic prior, since the deep representation is not fully constrained by dynamics. Our approach has a *weak* mechanistic prior since it is not encoded directly into the model, but rather it is learnt.

### 4.1    Nonlinear Ordinary Differential Equations

The first ODE model is the similar to the original application of latent force models (Lawrence et al., 2006): the biological process of transcriptional regulation. We validate additionally on experimentally-derived data from Barenco et al. (2006), where cancer cells were subject to ionising radiation and the concentration of mRNA was measured via microarray at different timepoints. The data contains transcript counts for five targets of the transcription factor p53 over seven timepoints and three replicates. We also consider the paired Lotka-Volterra equations, which govern predator-prey dynamics and exhibit periodic solutions.

**Latent Force Setup**    The transcriptional regulation task is defined by the ODE in Eq. 4, where the exact solution is only tractable when the response function is the identity. In this case, we set $G$ to the softplus function, $G(f(x)) = \log(1 + \exp(f(x)))$, to ensure positive protein abundance. We start by sampling parameters for Equation 4 from an empirical distribution of parameters learnt by running the *Alfi* (Moss

Table 1: Comparison to baseline models for the transcriptional regulation ODE and the the reaction diffusion PDE. For the DKLFM, we train on a dataset of 256 and 384 tasks for the ODE and PDE tasks respectively. *Alfi* and *DeepLFM* optimise each task independently. Results are averaged over 20 instances. DKLFM-a is an ablation the common component of the deep kernel is removed by setting $NN_c$ to the identity. DKLFM-b is an ablation of the Fourier embeddings, replacing them with a 4-layer MLP. Training was on an NVIDIA GeForce RTX 4090 GPU. Due to the differentiable PDE-solving package used by *Alfi* not being GPU-compatible, the PDE tasks were fit on an AMD Ryzen 5600x CPU. NLL is the negative log-likelihood, and the time column corresponds to the inference time per-task.

| TASK | MODEL | MSE ↓ (LATENT) | MSE ↓ (OUTPUT) | NLL ↓ | TIME (s) ↓ | MECHANISTIC |
|------|-------|----------------|----------------|-------|------------|-------------|
| ODE | Alfi (exact) | 0.117 | 0.0155 | $-1.29$ | 3.27 | Strong |
| ODE | DeepLFM | - | 0.0332 | 1.42 | 12.6 | Mid |
| ODE | DKLFM | **0.108** | **0.0028** | **$-2.39$** | $\mathbf{1.18 \times 10^{-2}}$ | Weak |
| ODE | DKLFM-a | 0.898 | **0.0027** | 8.64 | $\mathbf{1.01 \times 10^{-2}}$ | Weak |
| ODE | DKLFM-b | 0.321 | 0.0075 | $-2.07$ | $\mathbf{9.21 \times 10^{-3}}$ | Weak |
| PDE | Alfi (approx) | **0.0886** | 0.0215 | $-0.727$ | $> 600$ | Strong |
| PDE | DeepLFM | - | 0.356 | 0.547 | 96.7 | Mid |
| PDE | DKLFM | 0.131 | $\mathbf{1.92 \times 10^{-7}}$ | **$-1.96$** | **0.0523** | Weak |
| PDE | DKLFM-a | 0.998 | $3.01 \times 10^{-6}$ | $-0.799$ | 0.0606 | Weak |

et al., 2021) latent force inference package on the p53 network of genes experimentally measured by Barenco et al. (2006). Next, the latent force is sampled from a GP prior with RBF kernel, and the ODE is solved numerically using any differentiable solver, yielding a single task.

The Lotka-Volterra tasks are defined by the equations:

$$\frac{\mathrm{d}u(x)}{\mathrm{d}x} = \alpha u(x) - \beta u(x)v(x) \qquad \frac{\mathrm{d}v(x)}{\mathrm{d}x} = \gamma u(x)v(x) - \delta v(x), \tag{11}$$

where $u(x)$ and $v(x)$ are prey and predator populations respectively as a function of time, with growth rates $\alpha$ and $\gamma$, and decay rates $\beta$ and $\delta$. We use a periodic kernel for $\kappa$ in Eq. 5 for this task in order to capture the periodic nature of the Lotka-Volterra solutions, thus improving temporal extrapolation. We assume that we seek to infer the predator concentration from the abundance of prey; i.e., we take the predator population to be the latent force. We simulated a dataset of Lotka-Volterra solutions corresponding to different sampled rates, $\alpha, \beta \sim \mathcal{U}(0.5, 1.5)$, using a 4th-order Runge-Kutta solver. This is a slightly different regime to the transcriptional regulation model, where a Gaussian process was sampled for each datapoint. In this case, we validate our model's ability to infer a latent force that was not explicitly generated by a Gaussian process.

In both cases, Gaussian-distributed random noise is added to the latent forces. We generate 500 instances and split into training, validation, and test tasks. Figure 3 demonstrates that DKLFM can infer distributions over latent forces for the task of transcriptional regulation. We then apply the model trained on the simulated dataset to a real microarray dataset from Barenco et al. (2006), and show our inferred transcription factor concentration alongside the unobserved ground truth in Figure 4a. Next, we demonstrate the intra-task extrapolation in Figure 4b, where the input has been extended into the past and future. Finally, in Table 1, we compare our results with closest models from the literature, finding lower errors and computation times.

## 4.2 Partial Differential Equations

PDE-based LFMs are the multivariate extension of ODEs and are significantly harder to solve. This is illustrated by the absence of a method capable of exactly solving all classes of PDEs. Numerical solvers typically operate on a mesh and thus suffer the curse of dimensionality. Here, demonstrating the flexibility of DKLFM, we fit reaction diffusion equations with a very moderate dataset of 384 low-resolution tasks. The test tasks can then be inferred at a much higher resolution compared to training time.

**Latent Force Setup** Reaction diffusion equations have many uses; in this paper we look at the biological process of Drosophila embryogenesis (formation of the fruit-fly embryo). The spatiotemporal RNA expression, $h(x, t)$ of gap genes is measured using a reaction diffusion PDE from López-Lopera et al. (2019):

$$\frac{\partial h(x, t)}{\partial t} = sf(x, t) - \lambda h(x, t) + d\frac{\partial^2 h(x, t)}{\partial x^2}. \tag{12}$$

Here, $s$ is the production rate of the driving mRNA, $f(x, t)$ is the latent force, $\lambda$ is the decay rate and $d$ is the diffusion rate. Notice that the latent force is 2-dimensional; DKLFM can take any multivariate input.

In order to simulate a dataset from Equation 12, we implemented the Green's function approximation from López-Lopera et al. (2019). This approximation gives the full covariance matrix, including cross-covariances between latent force and outputs, and is faster in this direction than *Alfi*. Since the joint covariance matrix is singular due to repeated inputs, sampling is implemented using the eigendecomposition rather than Cholesky. We generate 448 tasks in this fashion, taking less than two hours on an AMD Ryzen 5600x, of which 384 are used for training. From an empirical inspection of the gap gene dataset from Becker et al. (2013), we uniformly sampled production rates in the range $[0.2, 1.0]$, decay rates in the range $[0.01, 0.4]$, and diffusion rates in the range $[0.001, 0.1]$. For the latent force, we sampled the two lengthscales (corresponding to spatial and temporal dimensions) in the range $[0.1, 0.4]$ since both dimensions are normalised to $[0, 1]$.

We show that we can learn a general solution operator for PDE tasks in Figure 5, invariant to input resolution. In Table 1, we show how our framework compares against single-instance models. *Alfi* tends to obtain very accurate results due to backpropagating the loss through the solver. However, DKLFM beats *Alfi* in output MSE with a greatly reduced computational burden, and using a modest dataset of 384 tasks. DKLFM also achieves a competitive error for the latent function.

## 4.3 Performance and Approximation Cost

The utility of LFMs for large scientific datasets is limited by their lengthy training times. In genomics, a realistic scenario is where a bioinformatician will want to train an LFM on several thousand genes; for example, RNA velocity (La Manno et al., 2018) solves a splicing kinetics ODE on the entire human genome. This limits the use of the available approximations. Moreover, the Gaussian joint distribution used in this work is a model assumption needing verification. We therefore qualitatively demonstrate the uncertainty in our approximated posterior predictive distribution using quantile-quantile plots.

Our framework, however, solves many LFMs simultaneously rather than optimising a single instance. An analysis of the relationship between error and training set size is therefore key to finding the point our error rate drops to the level of solving an individual instance. If this point is less than or similar to the number of instances in a typical use-case, then it is computationally preferable to generate a simulated dataset of this size rather than to train individual LFMs. This is because the training dataset required to reach this performance becomes smaller than the evaluation set, making it more time-efficient to use the DKLFM. For this study, we compare against *Alfi*, an accurate nonlinear LFM approximation defined in Moss et al. (2021). We chose the ODE task, since the PDE solver in *Alfi* is too computationally intensive for this comparison. In Figure 6, we confirm our hypothesis by plotting the MSE versus dataset size for our model, and horizontal lines are the mean MSE for *Alfi* over a subset of 64 tasks.

In order to analyse the posterior uncertainty of DKLFM, we use quantile-quantile plots showing samples from our posterior compared with real, simulated samples. We do this for three different tasks: 1. the nonlinear transcriptional regulation ODE model in Section 4.1; 2) a logistic growth model with a sinusoidal nonlinearity; and 3) the reaction-diffusion experiment from Section 4.2. The logistic growth model measures the increase of a resource, for example a population or a plant, as it reaches its maximum value. The rate of increase in size or quantity of the resource at time $\boldsymbol{x} \in \mathbb{R}^1$, $h(\boldsymbol{x})$, is expressed with the ODE

$$\frac{\mathrm{d}h(\boldsymbol{x})}{\mathrm{d}\boldsymbol{x}} = rh(\boldsymbol{x})\left(1 - \frac{h(\boldsymbol{x})}{K}\right) + \beta G(f(\boldsymbol{x})) \tag{13}$$

where $r$ is the growth rate, $K$ is the carrying capacity (maximum resource size), $\beta$ is the response factor to the temperature, $f(\boldsymbol{x})$, and $G$ is the cosine function. This equation was chosen since it can result in a

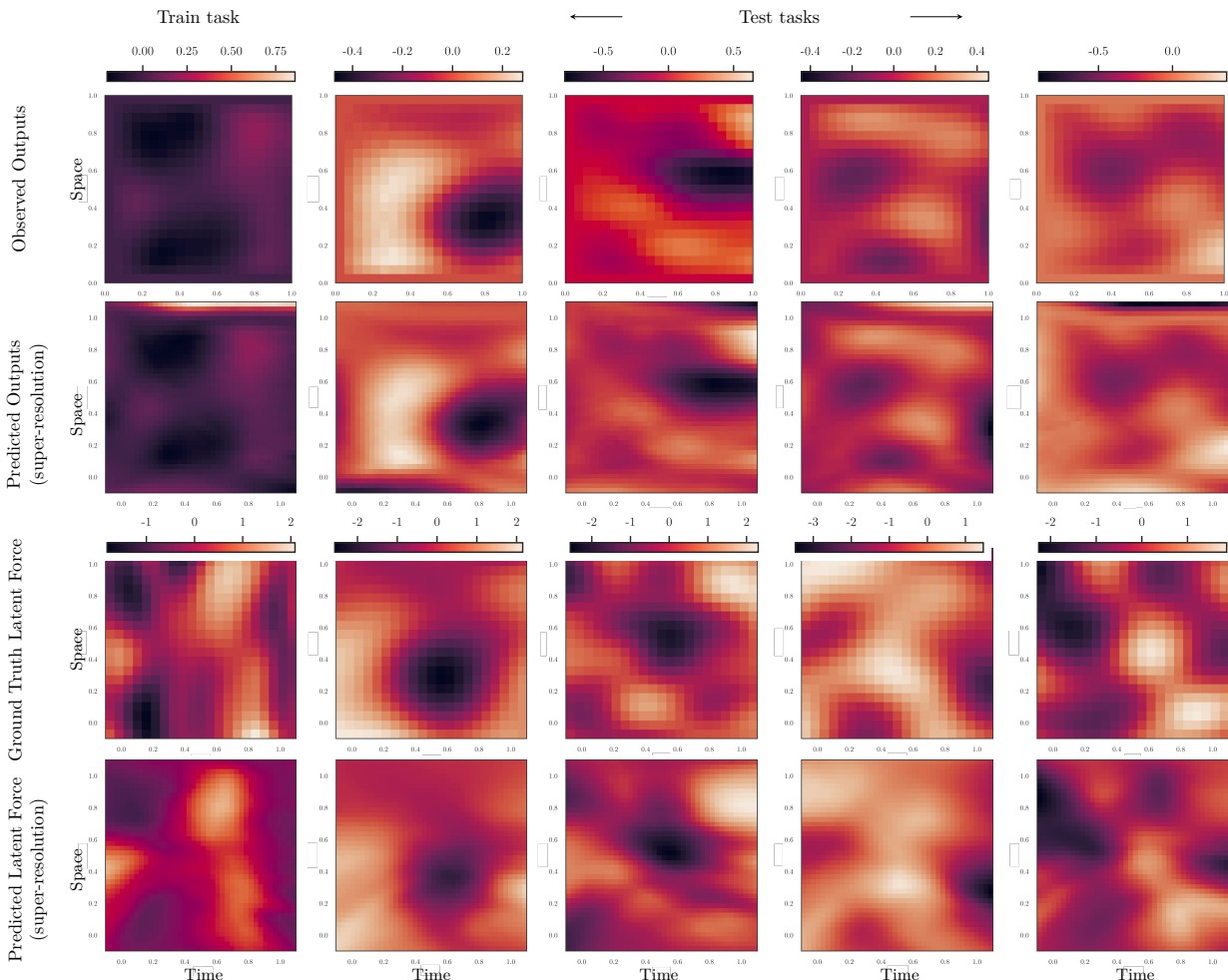

Figure 5: DKLFM trained on a synthetic reaction diffusion dataset. The first column is a training example and the next four are test cases, where the latent force is not observed. The embedding size was increased to 96 to account for the increase in dimensionality. The model was trained with a $21 \times 21$ spatiotemporal grid. At prediction time, $40 \times 40$ grid was used to illustrate the super-resolution capability. Each pair of plots vertically shares the same colorbar to enforce the same scale and accurately demonstrate inference accuracy.

multi-modal distribution due to the sinusoidal response to temperature. The QQ-plots shown in Figure 7 demonstrate that our assumption that the output function can be distributed as a multivariate Gaussian leads to robust uncertainty quantification. Even for the edge case of the logistic growth model, the associated QQ-plot exhibits a close fit to the unimodal Gaussian distribution.

## 5 Related Work

Differential equation-based inference in dynamical systems with Gaussian processes was introduced in Lawrence et al. (2006) and Alvarez et al. (2009). These approaches derive the kernel functions by solving the convolution integral of a base kernel with a linear operator corresponding to the ODE solution. When the dynamics becomes nonlinear, the Laplace approximation was used for the marginal likelihood. The benefit of this approach is that it is entirely non-parametric and enforces strong mechanistic behaviour. The primary issue is the requirement of analytically solving the specific ODE, which is not possible for some equations. Moreover, the first and second derivatives of the nonlinearity are also required. Also in line with

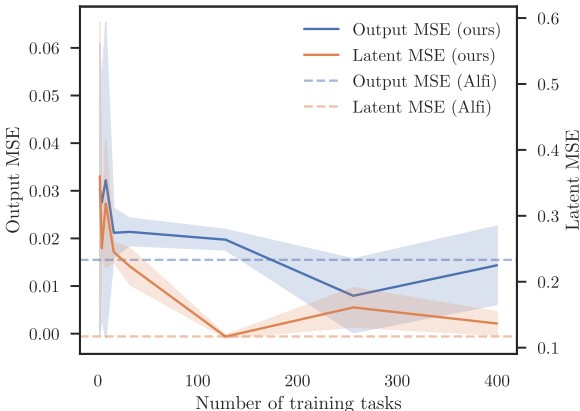

Figure 6: We plot MSEs against training dataset size for the genetic regulation LFM. This demonstrates the point at which it becomes more economical to use DKLFM rather than single-task models. At around 200 instances, the performance of DKLFM matches that of a single-task model optimised by Alfi.

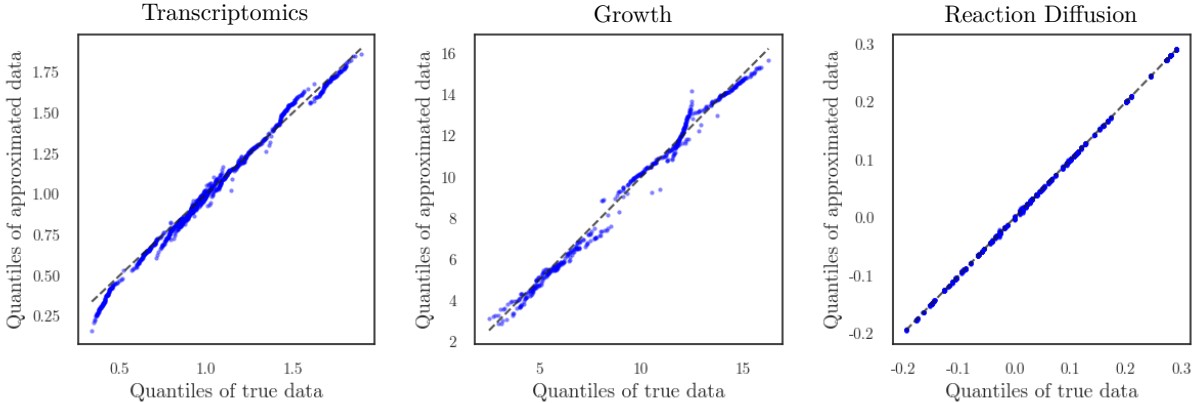

Figure 7: Quantile-quantile plots demonstrating the impact of our model approximation of a multivariate Gaussian join distribution in Section 3.1. We plot sorted samples at various timepoints linearly spaced across the input domain for three tasks. The transcriptomics task exhibits some deviation from an exact quantile fit due to the nonlinear transformation of the latent force. The reaction diffusion task involves only linear transformations of a Gaussian process resulting in a very tight close quantile fit. The logistic growth model can result in a multimodal (and therefore non-Gaussian) distribution.

this method is the switching LFM (Alvarez et al., 2010a), which segments time and switches between a number of latent forces, with only a single force being active in each interval.

Filtering approaches have also been investigated (Hartikainen & Sarkka, 2012; Särkkä et al., 2018; Ward et al., 2020). They typically employ a state-space model for approximating the posterior of a non-linear LFM. For example, Ward et al. (2020) use autoregressive flows to construct a joint density of the state using variational inference, bypassing complex kernel derivations. This results in a flexible model, scalable to multiple latent forces and output functions. However, the model exhibits the over-confidence prevalent in such black-box variational approaches.

Alfi (Moss et al., 2021) avoids the complex derivations of kernel functions by sampling the latent force from the GP prior and gradient matching to pre-estimate reasonable parameters. An ODE or PDE solver is then used to fine-tune the parameters with the solution of the equations. Backpropagating through a solver is far too computationally intensive for a multi-task setting, rendering this approach impractical for large datasets.

McDonald & Álvarez (2021) tackles non-linear and non-stationary dynamics by constructing a deep GP (Damianou & Lawrence, 2013). At each layer, an RBF kernel is convolved with the Green's function of the ODE. This deep representation enables the modeling of a wider range of tasks than a standard LFM. It is, however, not directly applicable to PDEs or to a multi-task setting.

There is also an adjacent area of research applying deep kernels for dynamical modelling. For example, Botteghi et al. (2022) denoise observations of the output functions via variational autoencoders, and learn a neural network representation of dynamics. In addition, Chen et al. (2022) propose a meta-learning framework for deep kernels consisting of a bilevel optimisation problem. The inner level consists of learning a subset of parameters which are adapted on individual tasks. At the outer level, the remaining subset of parameters are trained to yield the best loss on average over training tasks after the inner level is optimised. These approaches do not, however, infer latent forces. In this work, we treat the meta-network as an entirely separate component which is trained end-to-end with the deep kernel network.

## 6 Conclusion

We have introduced a novel framework for latent force models by leveraging the expressive power of deep kernels combined with a learned task representation. Where standard LFMs require an optimisation loop to find kernel parameters, our approach only requires GP conditioning on observations at prediction time, enabling extremely fast latent force inference. Specifically, this involves inverting a $T \times T$ matrix with $O(T^2)$ computational complexity. If the input size is too large, a technique such as variational inducing points reduces the computational complexity to $O(TM^2)$ with $M$ inducing points. DKLFM is therefore an *exact inference* probabilistic model: the first of its kind for learning the approximate solution operator for an arbitrary nonlinear LFM. We achieve this by learning a deep kernel corresponding to the differential equation by training on a simulated dataset of tasks. The embedding of each task's observations are interpreted as the task representation, containing information such as rate parameters. At test time, this representation is combined with an arbitrary input—possibly of higher resolution to the training data—in order to compute the latent forces using the Gaussian process conditioned on observations.

The inference performance of DKLFM is reliant on learning a good cross-covariance between latent forces and observations. We hypothesise that this is why this model does not exhibit the tendency for over-confidence away from training data in predictions commonly found in deep kernel learning and related approaches. Since we learn the same deep kernel over a dataset of tasks, this has proved not to be an issue. We demonstrate in two ablations the importance of both the common component of the deep kernel, $NN_c$, as well as the Fourier embeddings in learning a good representation of the dynamics.

**Limitations** Speed is limited by generating the training dataset; however, this is easily parallelised. Moreover, as in our PDE experiment, the generative direction, i.e. going from a parameterisation to an instance is much faster than the inverse problem of inferring the latent force from the output functions. We envisage these models being used analogously to large language models (Shanahan, 2022), where a user can fine-tune a pretrained DKLFM to make latent force inferences on their dataset. This would be particularly useful for computationally intensive simulations.

The original LFM is *strongly mechanistic*, deriving the covariance function from strict dynamics equations. While this may be more robust in the presence of lots of noise, it is overly rigid for real-world tasks. Our approach is *weakly mechanistic*: dynamics are not imposed, but rather parametrically learnt from the data and paired with a nonparametric GP to condition on unseen data. As with AlphaFold (Jumper et al., 2021), combining biophysical priors and data-driven approaches may be more appropriate for complex problems.

Despite not being an issue for the examples covered in this paper, our model approximation can lead to incorrect confidence predictions. Due to modelling the joint as a multivariate Gaussian, and therefore $h$ as a GP, the DKLFM theoretically cannot model transformations of the latent force that lead to non-Gaussian distributions over the output, $h$, such as those with multiple modes. However, Section 4.3 shows we still obtain close quantile fits even for a multimodal distribution.

**Further work** DKLFM explicitly treats the uncertainty in the latent forces and output functions. In an active learning context, we can query input points where the output function or latent forces have high-uncertainty. This enables experiment design, for example to determine an appropriate coarseness for a time-course experiment. Furthermore, we have currently considered one latent force per task. Multiple forces lead to identifiability issues, where many combinations of latent forces would solve the same LFM.

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

# A    Appendix

## A.1    Pseudocode

---

**Algorithm 1** Training procedure of DKLFM

---

**for** each task $n$ **do**
    $\boldsymbol{r}_n \leftarrow \mathrm{emb}(\boldsymbol{x}_n, \boldsymbol{y}_n)$                       ▷ Compute representations
    $\boldsymbol{z}_n = \boldsymbol{r}_n \oplus \boldsymbol{x}_n$                            ▷ Concatenate inputs
    $\boldsymbol{K}_{ff} = \kappa\left(\mathrm{NN}_c(\mathrm{NN}_f(\boldsymbol{z}_n)), \mathrm{NN}_c(\mathrm{NN}_f(\boldsymbol{z}'_n))\right)$       ▷ Latent force covariance
    $\boldsymbol{K}_{fh} = \kappa\left(\mathrm{NN}_c(\mathrm{NN}_f(\boldsymbol{z}_n)), \mathrm{NN}_c(\mathrm{NN}_h(\boldsymbol{z}'_n))\right)$     ▷ Cross-function covariance
    $\boldsymbol{K}_{hh} = \kappa\left(\mathrm{NN}_c(\mathrm{NN}_h(\boldsymbol{z}_n)), \mathrm{NN}_c(\mathrm{NN}_h(\boldsymbol{z}'_n))\right)$        ▷ Output covariance
    $\boldsymbol{\mu}_{h|f} = \boldsymbol{\mu}_h + \boldsymbol{K}_{hf}\boldsymbol{K}_{ff}^{-1}\boldsymbol{f}$
    $\boldsymbol{K}_{h|f} = \boldsymbol{K}_{hh} - \boldsymbol{K}_{hf}\boldsymbol{K}_{ff}^{-1}\boldsymbol{K}_{fh}$
    $p(\boldsymbol{y}, \boldsymbol{f}) = \mathcal{N}(\boldsymbol{f} \,|\, \boldsymbol{\mu}_f, \boldsymbol{K}_{ff})\mathcal{N}(\boldsymbol{y} \,|\, \boldsymbol{\mu}_{h|f}, \boldsymbol{K}_{h|f} + \sigma^2 I)$
    backpropagate $-\log p(\boldsymbol{y}, \boldsymbol{f})$
**end for**

---

---

**Algorithm 2** DKLFM inference

---

**for** each task $n$ **do**
    $\boldsymbol{r}_n \leftarrow \mathrm{emb}(\boldsymbol{x}_n, \boldsymbol{y}_n)$                       ▷ Compute representations
    $\boldsymbol{z}_n = \boldsymbol{r}_n \oplus \boldsymbol{x}_n$                            ▷ Concatenate inputs
    $\boldsymbol{r}_n^* \leftarrow \mathrm{emb}(\boldsymbol{x}_n^*, \boldsymbol{y}_n^*)$
    $\boldsymbol{z}_n^* = \boldsymbol{r}_n \oplus \boldsymbol{x}_n^*$
    $\boldsymbol{K}_{hh} = \kappa\left(\mathrm{NN}_c(\mathrm{NN}_h(\boldsymbol{z}_n)), \mathrm{NN}_c(\mathrm{NN}_h(\boldsymbol{z}'_n))\right)$        ▷ Output covariance
    $\boldsymbol{K}_{y*h} = \kappa\left(\mathrm{NN}_c(\mathrm{NN}_h(\boldsymbol{z}_n^*)), \mathrm{NN}_c(\mathrm{NN}_h(\boldsymbol{z}'_n))\right)$       ▷ Output covariance
    $\boldsymbol{K}_{f*h} = \kappa\left(\mathrm{NN}_c(\mathrm{NN}_f(\boldsymbol{z}_n^*)), \mathrm{NN}_c(\mathrm{NN}_h(\boldsymbol{z}'_n))\right)$     ▷ Cross-function covariance
    $\boldsymbol{\mu}_{f*|y} = \boldsymbol{\mu}_{f*} + \boldsymbol{K}_{f*h}\boldsymbol{K}_{hh}^{-1}\boldsymbol{y}$
    $\boldsymbol{K}_{f*|y} = \boldsymbol{K}_{f*f*} - \boldsymbol{K}_{f*h}\boldsymbol{K}_{hh}^{-1}\boldsymbol{K}_{hf*}$
    $\boldsymbol{\mu}_{y*|y} = \boldsymbol{\mu}_{y*} + \boldsymbol{K}_{y*h}\boldsymbol{K}_{hh}^{-1}\boldsymbol{y}$
    $\boldsymbol{K}_{y*|y} = \boldsymbol{K}_{y*y*} - \boldsymbol{K}_{y*h}\boldsymbol{K}_{hh}^{-1}\boldsymbol{K}_{hy*}$
**end for**
**return** $\mathcal{N}(\boldsymbol{\mu}_{f*|y}, \boldsymbol{K}_{f*|y}), \mathcal{N}(\boldsymbol{\mu}_{y*|y}, \boldsymbol{K}_{y*|y})$

---

