# OpenReview forum: "Deep Kernel Learning of Nonlinear Latent Force Models"
_TMLR — Accepted by TMLR_

### Review · Reviewer_pBCA · 2024-04-23

**Summary Of Contributions:**

This paper proposes to couple kernel input-warping to the LFM model, which interpolates PDE solutions from an observed dataset of related PDEs. The main contribution is adding a neural network inside the kernels to enrich the otherwise quite rigid kernel model. “Deep kernels” are known to be a useful approach, and adding it to LFM models is a good idea and has clear potential for dramatic empirical improvements.

**Audience:**

No

**Broader Impact Concerns:**

No issues

**Claims And Evidence:**

No

**Requested Changes:**

To be accepted, the paper needs to

(i) Define its claims, open problem, and contribution rigorously; and define the applications or usecases of the approach [demonstrate interest and claims]

(ii) Improve the clarity of the mathematical presentation

(iii) Add ablations to explain how the contributions (embedding + NN_) improve [claim evidence]

I'm looking forward to author clarifications and revisions. I hope my critique below is not taken as malice: I was genuinely confused.


Running comments reading the paper for first time

- Sec 2 LFM description is confusing. It is not clear what is the nature of x (do we have T scalar values or one T-dimensional vector; is this a multi-output system?), what is PxT (is this T vectors of size P, or one matrix?), or what the force does (why it is T → T?). Is is not clear what is the \cal{D} or what is the differential over. It is unclear what kind of process this describes (is this PDE or ODE or something else?). It’s also unclear where the Dy goes or what we do with it, or why do we care about it. If we do regression, why do we care about differentials? The paper would become much clearer by starting with a definition of ODE/PDE and defining that the dy/dt = h( t, x, f(x,t) ) is the main interest in this paper, and then tying all subsequent math to this master equation. Most of my confusions below stem from the discrepancy between the overly "static" probabilistic perspective and the PDE presentation of the experiments, and not defining the PDE setting in the beginning of the paper.
- I’m confused why the latent force is scalar in 3.1 even though we have P outputs. Isn’t it quite limiting to have scalar force? Can’t we have vector force?
- It’s also a bit confusing what the X,Y,f,h represent. How should we understand these? How come we are observing the latent forces? It would be good to have an example to concretise what these all mean in some simple usecase. These do not connect with PDEs currently.
- It’s not clear what “inference time” means. What do you “infer”? Is this network predictions or parameter posteriors, or something else?
- I think the above means that you don’t use the observed latents during training. But why not? Surely you want to use all the observations to train the model.
- It’s not clear what “h” means or represents. What is its domain and image? What does it do?
- What is K_ff or K_hf? Please define these kernels and their domains. I think these are cross-function kernels, but this is not made clear.
- It’s really strange the the model definition section does not define what kind of function you are learning. Can you define something like y = h( f(x) ) or maybe it’s y = h(x, f(x)) or maybe y = h(x) + f(x). It’s also not clear if this is regression or time-series fitting or something else. What task are you solving? I’m not really sure since the paper does not define what kind of function they estimate. I also can’t see where is the differential D or g() here. Did we forget about them? We just talked about them in eq 3. If they don’t appear in the paper, why did we introduce them? [Looking back, I think the system in this paper is dy/dt = h(t,x,f..), and I did not guess this from first read]
- mu_f is undefined in eq 4. Is this a prior? (But prior is m_f…). Same issue with mu_h, what is it?
- What does K_hf mean in eq 5? I don’t understand what kernel is this, or between what. Is the idea that h and f are measured at different input points? Isn’t this a cross-function kernel? Please be rigorous and give some exposition to this.
- Earlier it was stated that you don’t use f during inference time, but eq 5 seems to use it. Can you explain?
- Why do you want to infer the f* at testing time? What does f mean? Why do we care about? I thought that the idea is to estimate the output y during testing time (or perhaps the h). What kind of predictions do we want to do, and can you give some example usecase that would concretise these concepts?
- In eq 4 we define the GP h|f. Doesn’t this lead to the h and f being the same function? If you evaluate eq 5 for h at the same locations as the f, you would just get h=f+mu, ie equal up to a constant. I wonder what is the underlying assumption on how h and f relate? I don’t really see how can you cross-interpolate between two function like this: this looks like just interpolating the function f to new locations. I think this already comes down to the (joint) equation, which looks like a joint of a single GP, but I guess it’s some kind of cross-joint between two GPs. I’m not convinced this is correct or you can do this if you mean to have two different GPs. I don’t think two different GPs are jointly Gaussian. (h has not been defined so I’m not sure what kind of object it is). Is the idea that h=f? I’m quite confused.
- In sec 3.2. it is mentioned that we are interested in “reactions” and “decay rates”. Ok, so it seems that we are now talking about some kind of physico-chemical temporal systems, perhaps ODEs or SDEs or PDEs. This was not clear from earlier presentation, where nothing like this has been defined. The sec 3.1. does not have time anywhere!! You should link all sections of the paper so that they all describe the temporality and differentiality of the system, and hopefully start from the du(x,t)/dt definitions.
- It’s unclear what the tasks are. I guess we are now talking about some chemical reaction systems, so are different tasks different chemical systems with different parameters or different initialisation, or both? Please be rigorous.
- Earlier m_f was defined as prior for f. I guess that the inputs are x. Now we have m_f(z). Since we concatenate, the z is larger in dimension than x. I think you are redefining m_f here. Please make this explicit.
- Why is K_fh in eq 7 bold and capital? Is this a matrix-variate kernel, or perhaps a kernel matrix? It does not look like it: I think it’s just a scalar kernel value. The notation needs to match the tensor sizes (or please define what the notation means).
- I’m quite confused what $h$ and $f$ mean in the context of eq 7. Suddenly we map the same input z to both h and f through the two networks. Err… what? I thought that h takes as input the f, which comes from x. Surely you can’t just skip this and go directly x → h and x → f. I’m quite confused what these things mean, or how should we interpret them. How should we conceptualise this?
- I’m quite confused what the embeddings NN_f and NN_h and NN_c mean or what is their motivation. What are we trying to achieve here? Is this a feature extraction, or dimensionality reduction? What are the input and output sizes of these networks? Why do we have the NN_c? Isn’t it odd to map two different embedding with same network?
- I also now realise that I’m not sure what x_n means. I think it’s just R^D vector, but it might also be something else, since the discussion kinda implies towards set embeddings.
- I’m quite confused of the discussion around embeddings and transformer. I’m not sure what we are achieve here, or what is the motivation for this stuff, or what is our goal or problem.
- I don’t see where the “emb” goes in the equations. Currently it’s nowhere. I don’t understand why the emb takes as input PBTD. Why this choice? I think Y is of dim (B,T,P) and X is of dim (B,T,D) and f is of dim (B,T) How do we then go from these things to the (P,B,T,D)? What is the input to the emb? Why is the output (P,B,L)? Why not (P,L) or (B,L)?
- The fourier description is way too vague, and I have no idea where these things go in the equations, or what terms are we talking about. I guess this is about NN_f = Fourier or NN_h = Fourier? Or is this about emb=fourier? Likewise for transformer: not clear what parts of the model it operates over.
- The discussion here mentions meshing multiple times. I wonder where does this go in the equations. Is this just about the choice of X? But isn’t X given us by the data?
- It is suddenly mentioned that you are mesh-invariant. Where did this come from? Is this just about having a GP? Isn’t every GP mesh-invariant? I’m not sure what kind of argument you are making here.
- The paper mentions cross-covariance, which was not defined earlier. Can you make this more rigorous: what is the cross part?
- The paper also mentions that kernels come from some PDE definitions. Can you define this? Currently all kernels seem to be undefined.
- I’m not sure where the NN : R^L1 → R^L2 go. Does this replace NN_f, NN_c, NN_h? But the equation (periodic) still has these separate..
- Ok, so looking at eq 8 it seems that y is a function of time. I would not have guessed this based on the basic definitions in sec 3. In Sec 3 the Y (and other stuff) is represented like doing standard regression, and the notion of temporality is not present. Here we then suddenly have full-blown ODEs emerging. I also would not have guessed that the task refers to having different forcings. Please define the research setting earlier in the paper.
- What is the h in the example systems? What is y, h, f in the LV equations? It would be good to define all of these (eg. in a table). What do you observe in the example systems, what is data?
- The LV mentions a periodic kernel. But on what or between what?
- I don’t understand what are the different tasks in the LV. Is this different forcing choices or different initialisations or something else?
- I’m not sure what you are doing in the experiments. For instance in the LV, what do you learn? I think you learn the forcings, but isn’t this the task? Do you want to learn the time derivative (but the probabilistic equations don’t have time derivatives…)? Or do you want to just interpolate the observed states? Do you run some ODE forward in time? Which one? I’m really struggling to connect the underlying system, the probabilistic model, the data, the assumptions you make, the knowns/unknowns, and the goal you want to achieve.
- I’m also not sure what you are comparing your method against. What problem do you want to solve? Which methods do you want to improve? Why do you compare against these two methods?
- I think your learning task is to learn the forcing f(t) while also ensuring the solution y(t) is also accurate. Why do you want to do this? Can you demonstrate why this is an interesting and significant problem? What are the applications of this?
- In the PDE problem you generate many system solutions under different forcings (I assume). I think you are trying to learn the solution map by using data of related solutions. But you already know the true system (eq 10). If you already can generate 448 runs, why can’t you then generate the 1 test run you actually want to get without doing the 448 runs at all?
- Looking at fig 5, I think your approach is that you simply extrapolate the solution y_test(x,t) and forcings f_test(x,t) from a bunch of {y_train} and {f_train}, without running the PDE solver. [It would be great to define this as an equation.] I think this type of approach can be useful, but it requires likely a lot of observations of related systems, and will fail unless your dataset contains examples that had similar forcing to f_test and similar initial states to y_test. You also require the knowledge of the true system equation. Can you then explain what is the benefit or usecase for this approach? Why or when should we use this?

**Strengths And Weaknesses:**

S: The main idea is sensible and strong: adding DNNs to LFM is a great idea!

S: The results are good

W: The paper is written for LFM audience, and I struggled a lot to understand the paper due to numerous ambiguities, implicit assumptions and sloppiness in the mathematical presentation, and also in terms of claims, research setting, motivation and problem description. The research setting is barely defined (ODEs are first described half-way in the paper), the probabilistic model needs more exposition, and the embedding part is barely even defined. The presentation clarity needs to be significantly improved and clarified such that the method and contributions are un-ambiguous, precise and rigorous. Otherwise the paper is well written and flows well.

W: The results indicate the the system can learn or solve simple ODEs, and has competetive performance two few baselines. The main issue is no ablation comparison to a standard LFM without embeddings or deep kernels, and the paper needs to show how these two contributions improve.

---

> ### Comment · Reviewer_pBCA · 2024-05-29
>
> Some further comments
>
> - Eq 3 dh/dx is a Jacobian of size PxD, which surely is not the intention. If this is an ODE, it should be of size Px1. Text implies that eq 3 is an ODE, but it’s not since it has multiple dependent variables. I don’t understand the difference between h(x) and \bf{h}.
> - A GP prior is supposed to capture “biological noise”. What is “biological noise”? Why would we want to capture it? I’m quite confused: I thought the latent force is some meaningful forcing instead of just noise. Can you clarify the setting?
> - Fig 1 does not contain h, despite h being the main variable of interest. Can you include all key terms in fig1?
> - In sec 3.1. the \bf{x} changes to non-bf x. Can you explain?
> - h(x) is said to be a time derivative of mRNA, but eq 4 gives the derivative of it. Is eq 4 then a second time derivative ddh/ddt? I’m quite lost.
> - Eq 4 has again h(x) and \bf{h}, which I’m unsure about. The dh(x) looks awfully like a derivative or change, but apparently \bf{d} is not a derivative operator but parameters. Perhaps consider using some other symbol so that it’s not confused with derivatives. h(x) is P-dim, so dh notation does not work. Maybe it should be d \circ h?
> - Eq 3 and 4 have total derivatives wrt dx. Can you clarify that this is correct instead of using a partial derivative?
> - h is declared as GP. Hmm… Can you do this? You just defined h(x) in eq 4, and surely this does not induce a GP in general. Can you clarify more precisely what you mean by h being a GP?
> - c_f is undefined
> - I’m again confused of the relationship between eq 4 differential equation, and the GP definition of eq (joint). How can you have both at the same time? Is eq 4 a constraint that the eq (joint) has to follow? If we have a non-linear G, the distribution of h is not Gaussian anymore [I believe], so the entire GP joint construction becomes invalid. You can still approximate this dependency with a Gaussian with some covariance, but it wont’ be exact. Can you clarify and make this more precise?
> - I can’t follow the task discussion in deep kernels. The model presented has no notion of multitasks, so I struggle to follow what happens here. Surely you would just learn the tasks separately; or you would need to somehow incorporate the multi-task nature in the main equations themselves (ie. eq 3 and 4). I also fail to follow where the r comes from.
> - I also don’t understand what “task” means in this paper. Are the tasks just iid observed trajectories from the same system, or is each task an observed trajectory from a different system? Eg. we could observe data from a yeast and a bacterial cell; or we could observed two trajectories from same bacterial cell; or from two different bacterial cells (etc). Can you clarify?
> - Shouldn’t G be part of eqs 6-10? Shouldn’t G be part of the f or mu_h in eq 7?
> - In experiments the G is nonlinear. Doesn’t this invalidate your entire model? Surely the model is not applicable for G since things are not Gaussian or joint anymore. Please make the role and effect of G explicit throughout the paper.

---

### Review · Reviewer_SrVB · 2024-05-10

**Summary Of Contributions:**

The paper proposes a new deep kernel method for learning latent force models, which is applicable to a wide array of problems that make use of differential equations.

**Audience:**

Yes

**Claims And Evidence:**

Yes

**Requested Changes:**

*Content*:
- (suggested, not critical) Offer more systematic discussion of the complexity and scalability of the proposed method compared to existing methods. It's mentioned that there's a speed up of 250x to 15000x, but that's on a few specific tasks, right? What do you expect the pattern to be more generally, and is the evidence these few applications, or theoretical results about complexity, or something else? The complexity of one part of the approach is briefly mentioned in the discussion, but it's not clear how this relates to the complexity of the other parts of the approach or how it compares to the existing methods
- (suggested, not critical) The abstract mentions that the proposed method is faster than existing methods, but the results also show that the performance is better, right? Could be worth also mentioning this in the abstract.

*Formatting*:
- remove the strange bars appearing in the right-hand margin on most pages
- fix the width of the table on p. 7
- the text in many of the figures is too small to be legible; remove any unnecessary text and make sure what remains is legible

**Strengths And Weaknesses:**

*Strengths*:
- paper is well written
- method is clearly explained, achieves good performance, and appears much more tractable/scalable than existing methods

---

### Review · Reviewer_gFy4 · 2024-07-28

**Summary Of Contributions:**

The article aims to improve over standard latent force models by bypassing intractability and enhancing scalability and performance.  The authors do so through deep kernel learning, which allows them to avoid solving the diff eqs or resorting to variational approximations. The proposal is validated experimentally against Alfi and DeepLFM on different datasets

**Audience:**

Yes

**Claims And Evidence:**

No

**Requested Changes:**

Please address my comments above. Also:
- Why incorporating DKL into LFM is a challenge, and thus, why should it be considered a contribution?
- How general is the proposed approach? Will it work for datasets of different nature to those considered here?
- Refer to the results in Table 1: what are the shortcomings (or the price to pay) for such an impressive performance figures?
- Does the introduction of DKL hinder interpretability in the proposed model? is that relevant in the considered applications?

**Strengths And Weaknesses:**

The paper motivates using deep kernel learning (DKL) to improve some shortcomings of LFM. The proposal is illustrated with a model of the derivative of the mRNA (eq.  4). This emphasis on models for genetics is also adopted in the experimental part.

Though the paper can contribute to the application of the DKL+LFM to genetics, there are some inconsistencies in the delivery of the contribution.

- Initially, the contribution is presented as a DKL extension to LFM. Theoretically/conceptually, I am not sure about the significance of such a contribution, given that the advantages of DKL are known in the general case and should apply to this particular case too
- The paper stated that the introduction of DKL will address issues with tractability, scalability and performance. However, this is tested only in a particular dataset type, meaning that the claim about the superiority of the proposed model, in the general case, is vaguely supported. I think that the general applicability of the proposal is debatable.
- The results (Table 1) show a dramatic improvement in terms of performance & computational cost. To give the reader more insight into these impressive results, perhaps the authors should elaborate on such differences and discuss the limitations of the proposal (besides the ones in line 313) regarding the cases/problems where its performance is not as appealing as that in Table 1
- From my lack of expertise in models/applications to genetics models, perhaps the contribution should be presented as an application of LFM to this community rather than a contribution to the LFM/GP community.

---

> ### Author Response · Authors · 2024-08-12
> **Response**
>
> Dear Reviewer,
>
> Thank you for your review of our work and your helpful comments.
>
> *Positioning*: We view the incorporation of DKL within the LFM paradigm as a contribution to the LFM field because of the scalability and flexibility afforded by the approach. In Table 1, we illustrate the significance of this by showing the time saving at inference time. We also show the extension of the model to PDEs in Fig 5. In Fig 6, we demonstrate that there is a trade-off between training data required and the accuracy of the model. While we agree that the advantages of DKL are known, its use within a multi-instance inverse problem is novel. We believe this renders our approach a contribution directly to the LFM community.
>
> *Challenges*: We note that the associated challenges relate to how the DKL framework is used and the requirement of task representation capable of learning a latent representation of dynamics.
>
> *Generality*: The approach defined in this paper is very general to linear and nonlinear ordinary and partial differential equations as long as we have examples of the forward solution to the equations.
>
> *Interpretability*: You raise a very interesting point regarding interpretability. To the best of our knowledge, the parameters in an LFM are usually not used in downstream analysis; instead, the LFM equations are used as an inductive bias to extract latent forces. The DKLFM does not produce interpretable parameters so can only be used in such cases, hence why we term it “weakly mechanistic”. However, future work could look at predicting the equations from the latent task representation.
>
> *Cost*: The price to pay for the performance in Table 1 consists of: i) lack of interpretability; ii) weak mechanistic constraints; and iii) requirement to create training data (see Figure 6). A longer discussion of this is in Section 6.
>
> Please could you elaborate on your query regarding applicability? Particularly, you note that we test only on one “dataset type”--what does this refer to?
>
> Thank you again for your time.
>
> Best wishes,
>
> The Authors

---

> > ### Comment · Reviewer_gFy4 · 2024-08-15
> >
> > Thanks for the clarifications.
> >
> > Regarding your last question, I meant that the proposed method is tested (only) on differential equations of genomic tasks. Thus, the *general applicability* of the method, particularly beyond that dataset, is not validated. How can it be argued that the method is not only successful for the dataset (type) considered?

---

> > > ### Author Response · Authors · 2024-08-24
> > > **Response**
> > >
> > > Thank you for clarifying.
> > >
> > > In this paper, we evaluate on:
> > > 1. Transcriptional regulation ODE
> > > 2. Spatio-temporal reaction-diffusion PDE
> > > 3. Logistic resource growth model
> > > 4. Lokta-Volterra predator-prey ODEs
> > >
> > > We treat each experiment independently, e.g. we do not apply the model trained on dataset 1 to test dataset 2. This is because we are trying to model the specific dynamics of the given problem--it would not make sense to mix and match models of different dynamics. However, we believe the breadth of these experiments is a good indication of general applicability.
> > >
> > > Many thanks

---

### Comment · Reviewer_Hk2J · 2024-04-19
**Absolutely not**

This isn't remotely close to my areas of expertise (robotics, 3D perception). Please unassign.

---

### Comment · Reviewer_pBCA · 2024-07-29
**Role of LFM**

The paper has been updated, and the clarity of the paper has improved significantly. I still have concerns.

In eq 4 we define an LFM, where the structure of h(x) is complicated due to the non-linear G(), and the ODE/PDE nature of the h-solution. Thus, the K_hh and K_hf are intractable. In the eq:(joint) you define that the relationship between h/f is simply Gaussian, but this is not true! The relationship under the LFM assumption is non-Gaussian, and if you re-define it as Gaussian the model won't correspond to an LFM anymore. It would be ok to make a Gaussian joint _approximation_, but in that case one would need to ensure some matching between the approximation and the true system (eg. by an ELBO). Thus, I think the model you construct has nothing to do with LFM's anymore, but it's simply some kind of GPLVM regression from latent GP to an observed values. I think this paper is ultimately a variant of the VGPDS or varGPLVM works of Lawrence and others. Can the authors argue why or how the model still retains its LFM nature?

The neural network kernels seem to only be defined for Khh and Khf, while I think the Kff is still just a SE kernel (or similar). This would be incorrect: one needs to use consistent kernelisation across (h,f). That is, all three kernel matrices need to share the same kernel.

I'm confused of the learning setting. What do you optimise and how? I'm guessing that you optimise for a point solution of $f$, and also for the point solution of the kernel parameters (which are...?) using eq 6. Can you clarify? I'm also confused in experiments how the LFM/GP are used together (or are they). Algorithm boxes for training and testing times would be useful.

The paper needs to improve the empirical verification of the claims. First, the paper needs to compare to vanilla LFM (or some suitable version of it); compare to a model where the neural networks are replaced by regular Gaussian kernel; and compare to some GPDS style models (Lawrence and others). Second, the paper should try to decompose how different parts of the model are affecting the results. This could mean doing ablations where you drop the Fourier embeddings, task embeddings, etc. one by one to see how the results evolve.

Looking forward to authors comments.

---

> ### Author Response · Authors · 2024-08-20
> **Response**
>
> Thank you for your comment and reviewing our changes. We are pleased that the clarity has been improved.
>
> **Model Approximation** Indeed, with a nonlinear G, the LFM solution becomes non-Gaussian, meaning approximations are required (line 85). If we refer to the first line of the marginal likelihood in Eq. 6, we can identify two routes: approximate inference or approximate modelling. Some approximate inference methods are the Laplace as used in Lawrence et al., 2006 or by deriving an ELBO, as you suggest, as in Moss et al., 2022. In this paper, however, we make a model approximation where we assume that the relationship between h and f is approximately Gaussian, and then show that, while mathematically incorrect for nonlinear $G$, this can be a reasonable assumption in practice in Figure 7 with QQ plots (see especially the sinusoidal growth model).
>
> **Comparison to vanilla LFM**: We have updated the paper to state that Alfi, a model to which we compare, reverts to the exact solution in the linear case. We cannot compare to an exact LFM for the PDE experiment as it is not known.
>
> The VGPDS is an approximate inference approach similar to the variational approximation in Moss et al., 2022. The DKLFM instead aims to model the cross-covariance with a deep kernel. Moreover, the VGPDS and varGPLVM does not assume a differential equation model, whereas the DKLFM does. We believe this places our approach within the LFM framework and body of literature. To that end, we did not make a comparison to the VGPDS models. Indeed, a novel model setup for GPLVMs and GPDSs would have to be constructed in order to compare.
>
> **Construction of Kff and consistent kernelisation** We have clarified in the paper how Kff is constructed in the requested algorithm box in Appendix 1, showing Kff is also involving the deep kernel. It is worth noting, if your comment on consistent kernelisation is understood correctly, that in the original LFM formulation the kernel is also not shared across (h, f), although they are both derived from the differential equations. In our case, we learn the deep kernel using a combination of shared neural networks and separate heads for h and f. Thereafter they share the same base kernel.
>
> Please see DKLFM-a in Table 1 for the ablation of the common kernel.
>
> Another ablation, DKLFM-b, has been added to Table 1 which replaces the Fourier embedding with an MLP. The paper has been updated with the results of the ODE task and we are running the ablation for the PDE task.
>
> Thank you again for your review and this helpful discussion.

---

### Decision · Action_Editor_xN3J · 2024-10-25

**Recommendation:** Accept as is

**Comment:**

The paper proposes to use deep kernel learnings (i.e. standard kernels over a learned embedding of inputs) in the context of latent force models. Despite its simplicity, the method is quite effective empirically, and the reviewers appreciated the improved manuscript. I recommend acceptance.

**Audience:**

Yes

**Claims And Evidence:**

Yes